# Investigation on the Contents of N^ε^-carboxymethyllysine, N^ε^-carboxyethyllysine, and N-nitrosamines in Commercial Sausages on the Chinese Market

**DOI:** 10.3390/foods12040724

**Published:** 2023-02-07

**Authors:** Wenjing Wang, Yafei Kou, Yanli Du, Mingyu Li, Jian Zhang, Aiping Yan, Jianhua Xie, Mingyue Shen

**Affiliations:** 1State Key Laboratory of Food Science and Technology, College of Food Science and Technology, Nanchang University, Nanchang 330047, China; 2Center of Analysis and Testing of Nanchang University, Nanchang University, Nanchang 330047, China

**Keywords:** sausages, advanced glycation end-products, N-nitrosamines, α-Dicarbonyls

## Abstract

Sausages are among the most popular meat products worldwide. However, some harmful products, such as advanced glycation end-products (AGEs) and N-nitrosamines (NAs), can be formed simultaneously during sausage processing. In this study, the contents of AGEs, NAs, α-dicarbonyls and the proximate composition were investigated in two kinds of commercial sausages (fermented sausages and cooked sausages) in the Chinese market. The correlations among them were further analyzed. The results showed that the fermented and cooked sausages had different in protein/fat contents and pH/thiobarbituric acid reactive substance values due to their different processing technologies and added ingredients. The N^ε^-carboxymethyllysine (CML) and N^ε^-carboxyethyllysine (CEL) concentrations varied from 3.67 to 46.11 mg/kg and from 5.89 to 52.32 mg/kg, respectively, and the NAs concentrations ranged from 1.35 to 15.88 µg/kg. The contents of some hazardous compounds, such as CML, N-nitrosodimethylamine, and N-nitrosopiperidine, were observed to be higher in the fermented sausages than in the cooked sausages. Moreover, levels of NAs in some sausage samples exceeded the limit of 10 µg/kg issued by the United States Department of Agriculture, suggesting that particular attention should be paid to mitigating NAs, especially in fermented sausages. The correlation analysis suggested that the levels of AGEs and NAs were not significantly correlated in both kinds of sausages.

## 1. Introduction

Sausages are processed meat products. They are popular worldwide and acquire distinctive flavors through various technologies, such as fermentation, ripening, cooking, and smoking. However, during the processing and storage of sausages, many harmful products, such as advanced glycation end-products (AGEs) and N-nitrosamines (NAs), are generated [1]. AGEs are formed primarily at the advanced stage of the Maillard reaction, which occurs between the carbonyl groups of reducing sugars and the amino groups of proteins [2]. In addition, AGEs are related to lipid oxidation and α-dicarbonyl compounds. AGEs consist of a complex class of heterogeneous compounds, including N^ε^-carboxymethyllysine (CML), N^ε^-carboxyethyllysine (CEL), pyrraline, glyoxal-lysine dimer, and methylglyoxal-lysine dimer [3]. Among these widely studied AGEs, CML and CEL are considered as the typical markers of AGEs for their characteristics of stability and universality presented in food [4,5,6]. AGEs can be divided into exogenous and endogenous AGEs according to their origins, and dietary AGEs as dominant sources of exogenous AGEs play an important role in the protracted buildup of AGEs in human body. Many animal studies have indicated that dietary AGEs are harmful [7,8,9]. Li et al. [8] suggested that the CML level is significantly higher in kidneys, heart, liver, and lungs for healthy rats with oral administration of pure CML, which might induce chronic diseases in these organs. On the other hand, low-dietary AGEs can decrease the biomarkers of inflammation and oxidative stress in patients and healthy human beings [6]. It has been reported that accumulation of AGEs may be associated with the development of many age-related diseases, such as diabetes, atherosclerosis, kidney disease, and Alzheimer’s disease [10]. 

During the processing of sausages, nitrite is commonly added, because it can play a key role in color protection, bacteriostasis, and shelf-life extension [11]. Despite its positive effect on the quality of processed meats, the addition of nitrite has raised serious concerns about food safety, as it can combine with amines to create carcinogenic nitrosamines. Amines that are involved in the nitrosation process are usually derived from the enzymatic or microbiological decarboxylation of amino acids and protein degradation products [12,13]. Among these amines, secondary amines are known as direct precursors of NAs, and primary amines are thought to be involved in the formation of NAs after conversion to secondary amines by reactions, such as cyclization and dimerization [13,14]. Nitrosamines are of several types; N-nitrosodimethylamine (NDMA) and N-nitrosodiethylamine (NDEA) are designated as Group 2A by the International Agency for Research on Cancer, and N-nitrosomethylethylamine (NMEA), N-nitrosodi-n-propylamine (NDPA), N-nitrosodibutylamine (NDBA), N-nitrosopiperidine (NPIP), N-nitrosomorpholine (NMOR), N-nitrosopyrrolidine (NPYR), and N-nitrosodiphenylamine (NDPhA) are classified under Group 2B [15].

At present, the concentrations of AGEs and NAs in foods can be determined using a variety of methods, such as liquid chromatography with tandem mass spectrometry (LC-MS/MS) [16,17], enzyme-linked immunosorbent assay (ELISA) [6], gas chromatography with tandem mass spectrometry (GC-MS/MS) [18,19], and high-performance liquid chromatography (HPLC) [1]. In comparison to the ELISA method, the LC-MS/MS method for AGEs is highly selective and can provide more accurate results in a complex food matrix [20]. Additionally, the results determined by LC-MS/MS are expressed as actual contents, while those in ELISA are arbitrary units. NAs are generally detected by GC-MS/MS in sausages [1]. Sannino et al. [21] employed GC-MS/MS for detection of NAs in meat products and found the method is sensitive with satisfactory parameters. 

Sausages are rich in proteins and lipids, and commonly nitrites are added as an ingredient, leading to the simultaneous formation of AGEs and NAs. The contents of AGEs or NAs in sausages have been reported by several studies [22,23,24,25,26]. For example, Yu et al. [24] investigated CML and CEL contents in eight groups of meat products and found that the average levels of CML and CEL in sausages were 60.9 and 65.01 mg/kg protein, respectively. De Mey et al. [22] collected 101 dry sausages in Belgium and found that the NAs contents in most sausages were below 5.5 µg/kg. Although these references have paid attention to levels of AGEs or NAs in meat products, these studies rarely evaluate the two typical harmful products (i.e., AGEs and NAs) in combination [1].

Therefore, this study aimed to examine the levels of AGEs (CML and CEL) and NAs (NDMA, NDEA, NMEA, NPYR, NMOR, NDPA, NPIP, NDBA, and NDPhA) in 21 kinds of sausages in the Chinese market. Furthermore, their proximate composition and α-dicarbonyls were analyzed to establish the correlation between harmful products and the analyzed chemical indices.

## 2. Materials and Methods

### 2.1. Reagents

AGEs standards including CML, CEL, d_4_-CML, d_4_-CEL and 2 internal standards of NAs (NDEA-d_10_ and NPYR-d_8_) were purchased from Toronto Research Chemicals Inc. (Toronto, ON, Canada). A mixture of NAs standard (NDMA, NDEA, NMEA, NDPA, NDBA, NPIP, NPYR, NMOR, and NDPhA), 3-deoxyglucosone (3-DG), quinoxaline, 2-methylquinoxaline, and *o*-phenylenediamine were obtained from Sigma-Aldrich (St. Louis, MO, USA). NDMA-d_6_ was purchased from O2si (Charleston, SC, USA). HPLC grade acetonitrile, methanol, and formic acid were bought from Merck Company (Darmstadt, Germany). HPLC grade dichloromethane was supplied by Shanghai Aladdin Biochemical Technology Co. (Shanghai, China). Bondesil octadecyl (C18) sorbent (40 µm) and Bondesil primary secondary amine (PSA) sorbent (40 µm) were acquired from Agilent Technologies (Santa Clara, CA, USA). Oasis MCX solid-phase extraction cartridges (6 cc, 150 mg) were obtained from Waters (Milford, MA, USA). Deionized water was provided by the Watson Group Ltd. (Hongkong, China). All other reagents were of analytical grade and were supplied by Sinopharm Chemical Reagent Co., Ltd. (Beijing, China).

### 2.2. Samples

A total of 21 sausages were obtained from Taobao online stores and local supermarkets in China. The samples were divided into two categories: fermented sausages (F1–F12), which were only fermented at low temperatures for an extended period, and cooked sausages (C1–C9), which were smoked or steamed at high temperatures. The samples were minced, ground in a blender and stored at −20 °C for further analysis.

### 2.3. Measurement of Moisture, Protein, Fat Content, and pH

Moisture was determined using the method of the Association of Official Analytical Chemists (AOAC) [27]. In brief, 2 g of sausage was dried at 105 °C to a constant weight in an oven. The protein content was estimated with the Kjeldahl method [27]. Briefly, 0.5 g of sample was distilled and titrated on a Hanon K9860 Automatic Kjeldahl Analyzer (Jinan Hanon Instrument Co., Ltd., China) with a conversion coefficient of 6.25. Fat content was measured by Soxhlet extraction according to the method of the AOAC [27]. The pH value was determined by following the method described by Xiao et al. [23]. Samples (1 g) were vortexed and homogenized with 9 mL of ultrapure water, and the filtrate was employed to estimate the pH value by using a pH meter (Mettler Toledo, Columbus, OH, USA).

### 2.4. Determination of CML and CEL in Sausages

The levels of CML and CEL in the sausages were determined using the method reported by Li et al. [28]. In brief, a 0.5 g sample was mixed three times with 5 mL of hexane and centrifuged for 5 min at 4800 r/min, and the hexane layer was discarded to remove fats. Then, the residue was dried with nitrogen gas, and the sample was reduced with 1 mL of sodium borohydride (1 M in 0.1 M NaOH) and 1.5 mL of borate buffer solution (0.2 M, pH 9.2) overnight at 4 °C. Afterwards, hydrochloric acid (12 M, 2.5 mL) was added, and the samples were hydrolyzed overnight at 110 °C for 24 h. The hydrolysate was filtered into a 25 mL volumetric flask, and the volume was fixed with ultrapure water. Next, 1 mL of the sample solution was pipetted and blow-dried with nitrogen gas, and the dried sample was reconstituted with 1 mL of water and spiked with 100 µL of internal standard solutions (2 µg/mL, d_4_-CML and d_4_-CEL) for solid-phase extraction (SPE). The Oasis MCX cartridge was preconditioned with 3 mL of methanol and 3 mL of water. Then, the reconstituted sample solution was loaded on a cartridge and washed with 3 mL of water and 3 mL of methanol. Finally, the target analyte was eluted with 5 mL of methanol/ammonia (95:5, *v/v*). The eluate was dried completely with nitrogen gas, resolved in 1 mL of ultrapure water and filtered through a 0.22 µm membrane for LC-MS/MS analysis.

The quantification of CML and CEL was performed using an Agilent 1290 Series high-performance liquid chromatography system (HPLC 1290) with a 6460 triple quadrupole (QqQ) mass spectrometer (Agilent Technologies Inc., Santa Clara, CA, USA). The analytes were chromatographically separated on a Synergi Hydro-RP 80Å LC column (250 mm × 2 mm, 4 µm, Phenomenex) at 25 °C. Notably, 0.1% formic acid aqueous solution (*v/v*) was used as the only mobile phase. The flow rate was 0.15 mL/min, and the injection volume was 2 µL. The QqQ mass spectrometer was operated in the positive electrospray ionization (ESI) mode with a multiple reaction monitoring (MRM) scan type. The mass parameters were as follows: gas temperature, 350 °C; gas flow rate, 10 L/min; sheath gas temperature, 350 °C; sheath gas flow rate, 9 L/min; nebulizer pressure, 25 psi; capillary voltage, 3.5 kV. The MRM parameters are presented in Appendix A. The limit of detection (LOD) and the limit of quantification (LOQ) of CML were 0.05 and 0.1 µg/g, respectively, and those of CEL were 0.01 and 0.03 µg/g, respectively. The coefficients of regression (R^2^) for CML and CEL were 0.9994, and their recovery rates were 90–108%. The chromatogram is shown in Appendix A. 

### 2.5. Determination of α-Dicarbonyls in Sausages

α-Dicarbonyls were determined using the method described by Hu et al. [29]. In brief, 0.5 g of the samples was combined with 3 mL of water and 3 mL of chloroform:methanol (2:1, *v/v*) and centrifuged for 10 min at 4800 r/min. The supernatant was transferred to another centrifuge tube, and the residue was re-extracted with 1 mL of water. Next, 200 µL of the combined supernatant was mixed with 400 µL of sodium phosphate buffer (0.1 M) and 400 µL of *o*-phenylenediamine solution (10 mg/mL) for derivatization. Then, the mixture was incubated at 25 °C for 20 h in the dark. Finally, the sample solution was filtered through a 0.22 µm membrane for LC-MS/MS analysis.

The quantification of α-dicarbonyls was performed on the same instrument with the same QqQ parameters as those given in Section 2.4. The separation of the derived α-dicarbonyls was accomplished on a Synergi Hydro-RP 80Å LC column (150 mm × 2 mm, 4 µm; Phenomenex) at 30 °C by using a 60% methanol and 40% formic acid aqueous solution (0.1%, *v/v*) as the mobile phase. The flow rate was 0.25 mL/min, and the injection volume was 2 µL. The MRM transition ions are presented in Appendix A.

### 2.6. Lipid Oxidation

The level of lipid oxidation in the sausages was assessed with thiobarbituric acid reactive substance (TBARS) values, which were measured using the method of Yu et al. [25]. In brief, 0.5 g of the samples was mixed with 3 mL of a thiobarbituric acid (TBA) solution (1% TBA solution with 0.075 mol/L NaOH) and 17 mL of a trichloroacetic acid (TCA)–HCl solution (2.5% TCA solution with 0.036 N HCl). The mixture was incubated at 100 °C in a water bath for 1 h after being filled with nitrogen gas. Exactly 5 mL of the supernatant was mixed with 5 mL of chloroform, and the sample was centrifuged for 10 min at 4800 r/min. The absorbance of the lower layer solution was acquired at λ = 532 nm. The result was calculated through the following equation:$$TBARS\;\left(mg/kg\;sample\right)=\left(A_{532}/W_{S}\right)\times 9.48,$$
where A_532_ is the absorbance of the sample solution at λ = 532 nm, Ws is the weight of the sample (g), and “9.48” is a constant.

### 2.7. Determination of NAs in Sausages

N-nitrosamines were detected with the method of Lehotay et al. [18] with some modifications. Specifically, 10 g of samples was mixed with 10 mL of 1% acetic acid in acetonitrile/water (1:1, *v/v*) and spiked with 100 µL of an internal standard solution (2 µg/mL, NDEA-d_10_, NPYR-d_8_, and NDMA-d_6_) in a 50 mL polypropylene centrifuge tube. Then, two Agilent BondElut ceramic stirring bars were added to the tube. The sample tube was sonicated for 15 min in an ice bath and stored for 30 min at −20 °C. Subsequently, 4 g of magnesium sulfate anhydrous (MgSO_4_) and 1 g of sodium chloride (NaCl) were added, followed by vigorous manual shaking for 30 s and vortexing for 1 min. Afterwards, the mixture was centrifuged at 8000 r/min (0 °C) for 10 min. Exactly 2 mL of the supernatant was transferred to a 10 mL centrifuge tube containing 150 mg of MgSO_4_, 50 mg of C18, and 25 mg of PSA. Next, 2 mL of hexane solution pre-saturated with acetonitrile was added to the tube, and samples were centrifuged at 8000 r/min (0 °C) for 5 min. The upper layer was loaded into a sodium sulfate SPE cartridge to remove possible water, after which the sample solution was dried to about 0.2 mL under a mild nitrogen stream at 30 °C and redissolved to a final volume of 1 mL with dichloromethane. The solution was filtered with a 0.22 µm membrane for GC-MS/MS analysis.

The analysis of NAs was performed on an Agilent 7890B GC coupled with a 7000D QqQ system (Agilent Technologies Inc., Santa Clara, CA, USA). Chromatographic separation was achieved on a DB-35MS capillary column (30 m × 0.25 mm × 0.25 µm; Agilent) with an injection volume of 2 µL. The carrier gas was helium at a constant flow of 1.5 mL/min. The oven program was set as follows: initially held at 60 °C for 2 min, then ramped to 180 °C at a rate of 10 °C /min, held for 1 min, increased to 250 °C at a rate of 20 °C /min and held for 8 min. The mass spectrometer was implemented in the electron ionization mode at an electron energy of 70 eV. The ion source and transfer line temperatures were 230 and 250 °C, respectively. The MRM mode was employed with the ions acquired in four segments, and the specific parameters are shown in Appendix A. The performance of the NAs is presented in Appendix A. The chromatogram is shown in Appendix A.

### 2.8. Statistical Analysis

All experiments were conducted in triplicate, and the results were expressed as mean ± standard deviation. Statistical analysis was performed through a nonparametric statistical test utilizing SPSS software (version 25.0; IBM Corp., Armonk, NY, USA) with the Kruskal–Wallis test. Correlation analysis was conducted using the Pearson correlation coefficients. The statistically significant levels of *p* < 0.05 and *p* < 0.01 were used.

## 3. Results and Discussion

### 3.1. Proximate Analysis of Sausage Samples

The sausages collected in the Chinese market were divided into two categories (fermented and cooked sausages) based primarily on processing techniques [30]. Fermented sausages are non-heated products that are fermented and matured under conditions of defined temperatures and humidity, leading to the occurrence of acidification and dehydration [31]. The raw materials of fermented sausages are usually minced meat with some ingredients, such as salts, nitrites, starter cultures, and spices. On the other hand, cooked sausages are usually formulated with water, starch, and soy protein, causing a decrease in fat content and an increase in moisture content compared with fermented sausages. Cooked sausages are heated products and go through some heat treatment, such as smoking, steaming, and cooking. Given that the two categories of sausages quite differ in their raw meat contents, methods of processing, and added ingredients, it is not surprising that their proximate compositions pH and the contents of moisture, protein, fat, and TBARS vary drastically. 

The proximate compositions of the collected commercial sausages are summarized in Appendix A and Figure 1. As shown in Appendix A, the moisture contents in the commercial sausages ranged from 14.1% to 68.9%. As visually displayed in Figure 1, high moisture contents with an average level of 63.7% were observed in the cooked sausages, whereas the average level for the fermented sausages was 30.0%. The fat levels ranged from 0.9% to 43.2% for the all sausages, and the average for the fermented sausages (34.6%) was higher than that for the cooked sausages (6.3%). The protein contents ranged from 10.0% to 29.2%, and the fermented sausages contained higher levels of protein compared to the cooked sausages with the average being 24.4% and 14.6%, respectively, except for Beidaihe Sausage (29.2%). The fermentation and long days of ripening were involved in manufacturing of fermented sausages, which contributed to lower moisture content and higher protein and fat content compared to the cooked sausages that were produced by heat treatment and short processing time.

The pH values of all sausages ranged from 4.9 to 6.9, and the average pH value of the fermented sausages (5.5) was lower than that of the cooked sausages (6.6). The fermentation process involving the addition of a starter culture is important for fermented sausages, which gives the sausages a tangy taste, creates an acidic environment in the meat and helps to prevent the growth of harmful bacteria. TBARS is an indicator to evaluate the degree of lipid oxidation [32]. In this study, the TBARS values ranged from 1.36 to 14.77 mg/kg. The average TBARS values of the fermented and cooked sausages were 8.14 and 2.62 mg/kg, respectively. The lipid oxidation in the sausages was influenced by several factors, such as fat contents, added ingredients (e.g., NaCl and nitrite), pH value, and storage time [31,33]. In general, the TBARS values of the fermented sausages were greater than those of the cooked sausages, which may be due to the higher fat content of the fermented sausages compared with that of the cooked sausages [33]. 

### 3.2. Determination of α-Dicarbonyls

α-Dicarbonyls (α-DCs) are intermediate products of the Maillard reaction and are highly active precursors of CML and CEL, in which 3-deoxyglucosone (3-DG), diacetyl (DA), glyoxal (GO), and methylglyoxal (MGO) are the most typical α-DCs [34]. The concentration of α-DCs in the collected sausages is presented in Appendix A and Figure 2. As shown in Figure 2, the fermented and cooked sausages did not differ significantly in their 3-DG, DA, MGO, and total detected α-DCs concentrations (*p* > 0.05). GO contents were observed to have a significant difference between the two categories of sausages (*p* < 0.05). 3-DG contents of the two sausages were highly variable with the distribution ranging from not detectable to 6.93 mg/kg. The most important factor affecting the formation of 3-DG is the amount of sugar, especially monosaccharides [35]. Generally, the sugars added to sausages are mainly glucose and/or sucrose, and their concentrations may be various based on different brands, leading to the highly variable content of 3-DG in sausages. DA contents in meat products have rarely been reported. In this study, DA levels detected in the sausages were low. They ranged from 0.08 to 0.76 mg/kg, and the average content was 0.22 mg/kg. MGO levels ranged within 0.21–3.06 mg/kg with an average content being 1.49 mg/kg. In 13 of the 21 samples, it was noted that MGO amounts were the highest among the detected α-DCs, which was consistent with the results of Maasen et al. [36], who found that the MGO amounts in fried meat products and salami were higher compared to GO and 3-DG amounts. In addition, GO amounts were higher in the fermented sausages (0.09–1.77 mg/kg) than in the cooked sausages (0.05–1.19 mg/kg), and the average values of the two categories were 1.08 and 0.36 mg/kg, respectively. This significant difference in GO content between the two sausage types might be due to the fermentation process in the fermented sausages. It is similar to the fact that high GO content in chocolate possibly originates from the fermentation step [36]. The total amounts of the detected α-DCs ranged from 1.11 to 8.35 mg/kg, which were lower than those in most food products, such as cookies, beer, and bread [35,36]. It was reported that creatin compounds in meat could serve as a scavenger of α-DCs, which might explain low amounts in sausages [37,38].

### 3.3. Determination of CML and CEL

The CML and CEL contents of the collected sausages are shown in Appendix A and Figure 3. As visually displayed in Figure 3, the CML contents were significantly higher in the fermented sausages than in the cooked sausages (*p* < 0.05). The CML contents in the fermented sausages ranged within 7.38–34.82 mg/kg, whereas those in the cooked sausages ranged within 3.67–46.11 mg/kg. The average levels in these two sausages were 15.61 and 11.68 mg/kg, respectively. This significant difference might result from different formation pathways in the two sausages. The pathways to generate AGEs were various including Maillard and oxidation reactions [39]. With regard to the Maillard reaction, GO was more abundant in the fermented sausages than in the cooked ones, and it acted as a direct precursor with lysine to form more CML. Meanwhile, the generation of AGEs was also related to lipid oxidation [25]. As shown in Figure 1, the TBARS values were high in fermented sausages, in which lipid oxidation proceeded intensely, contributing to the generation of CML and CEL. Specifically, the Maillard reaction rate was slow at low temperatures [40], and the oxidation reaction might play an important role in the formation of AGEs during low-temperature processing in fermented sausages [25,41]. In addition, CEL concentrations ranged within 5.89–52.32 mg/kg, and the average level was 16.67 mg/kg. No significant difference in CEL level was observed between the two types of sausages. These results were in accordance with the data of MGO, which was recognized as the precursor of CEL. In the fermented sausages, there was no significant difference between CML and CEL contents (*p* > 0.05), whereas in the cooked sausages, CEL contents were significantly higher than CML contents (*p* < 0.05). Sun et al. [42] reported that the activation energy in CEL formation (29.21 kJ/mol) was smaller than that in CML formation (61.01 kJ/mol) in ground beef. Therefore, CEL might be generated more easily at high temperatures compared to CML [16,43]. Generally, the processing temperature of cooked sausages is higher than that of fermented sausages, resulting in higher CEL levels than CML levels in cooked sausages. In the present study, the sum of CML and CEL concentrations ranged within 9.56–93.66 mg/kg, and the average level was 31.20 mg/kg; this result was in accordance with results reported by Yu et al. [24].

### 3.4. Determination of N-Nitrosamines

The levels of NAs in the collected sausages are given in Appendix A and summarized in Figure 4. NMEA, NDPA, and NMOR were not detected in any of the sausages. NDMA and NDEA are two primary NAs that were detected in most sausages [44]. As presented in Figure 4, no significant difference in NDEA between the fermented and cooked sausages was observed. NDEA contents ranged within 0.89–2.53 µg/kg with an average level of 1.55 µg/kg. However, NDMA levels were significantly different in the two sausages (*p* < 0.05), which was in agreement with the study of Scheeren et al. [19], who found that fermented sausages contain higher NDMA than cooked sausages. The NDMA contents in the fermented sausages ranged within 4.25–10.52 µg/kg, whereas the contents in the cooked sausages ranged from not detectable to 6.09 µg/kg. Their average levels were 6.96 and 3.22 µg/kg, respectively. The contents of NDMA and NDEA detected in the sausages were higher compared to the values in some previous studies [26,45]. For example, Niklas et al. [45] investigated the occurrence of nitrosamines in the Danish market and found that NDMA levels ranged between not detectable and 3.8 µg/kg in pork salami. Herrmann et al. [46] reported that the level of NDMA in salami was up to 7.2 µg/kg, which was comparable to our results. These differences in the contents of NDMA might be influenced by the recipe, processing technology, storage time, and other factors [13,47]. NDMA is liable to be produced in fat-rich meat than in lean meat [26,48]. The fermented sausages had more fat than the cooked sausages as shown in Appendix A and Figure 1, leading to higher NDMA contents in the fermented sausages than in the cooked ones. NPIP was detected only in the fermented sausages s, and its amount ranged from not detectable to 3.79 µg/kg. Spices, such as black pepper, are commonly used as ingredients in the manufacturing of fermented sausages. However, they contain piperidine, which might act as a precursor of NPIP, and might be responsible for the formation of NPIP [26,49]. NPYR was detected only in Shuanghui Corn Sausage, and the level reached 71.73 µg/kg. However, there was no clear explanation for this observation. Contrary to previous studies indicating that NPYR is frequently present in meat products [26,45], NPYR was not detected for most sausages in this study, probably due to the low processing temperature [26]. Previous research showed that NPYR frequently appears at temperatures higher than 200 °C [50]. NDBA and NDPhA levels between the fermented and cooked sausages were not significantly different. NDBA was detected in 43% of all selected sausages, and its content ranged from not detectable to 2.20 µg/kg with the average level being 0.40 µg/kg. NDPhA was found in all the sausages at low levels, which ranged from 0.08 to 2.07 µg/kg with an average content of 0.54 µg/kg. These results were consistent with those of previous studies, which reported that no or low levels of NDBA and NDPhA were observed in meat products [19,26,51]. The total detected NAs levels among all the sausages ranged from 1.35 to 15.88 µg/kg, except for Shuanghui Corn Sausage, and part of samples were above the limit of 10 µg/kg recommended by the US Department of Agriculture (USDA) [52]. These results suggested that the factors influencing the formation of NAs, such as meat quality, processing temperature, and storage conditions, should be controlled [13].

### 3.5. Correlation Analysis

The formation of AGEs and NAs occurs simultaneously during the processing and storage of sausages and are influenced by many factors, such as added ingredients, meat types, processing technology, and storage conditions. Therefore, the pathways involved in AGEs and NAs formation in sausages are notably complex. Correlation analysis can help to understand their mechanism better. In this study, correlation analyses were conducted separately due to the significant differences between fermented and cooked sausages in terms of their proximate composition and processing technology. As seen in Figure 5, a significant positive correlation was found between CML and CEL in the cooked sausages, but no significant correlation was observed in the fermented sausages, suggesting that CML and CEL formation pathways in fermented sausages are different and in cooked sausages these pathways are closely associated. This difference might result from different processing technologies and proximate components. Furthermore, CEL was also observed to be positively correlated with fat content and pH in the fermented sausages, whereas CML and CEL showed a significant positive correlation with protein in the cooked sausages, indicating that the formations of AGEs in the fermented and cooked sausages are primarily due to lipid oxidation and the Maillard reaction, respectively. Li et al. [53] also reported that during storage at low temperatures, the Schiff base involved in the Maillard reaction is weakly correlated with AGEs, but a strong correlation is exhibited at high temperatures in surimi products. Moreover, CEL had a significant positive correlation with 3-DG in the fermented sausages, whereas AGEs were not significantly correlated with α-DCs in the cooked sausages, further demonstrating the different formation mechanisms in the two sausages. For α-DCs, a significant negative correlation was observed between 3-DG and GO in the fermented sausages, and a close correlation was observed among α-DCs in the cooked sausages, implying that the formation pathways of α-DCs in fermented sausages might be different whereas those in cooked sausages might be similar.

Some NAs, such as NDMA/NPIP and NDEA/NDBA in the fermented sausages and NDMA/NDPhA and NPYR/NDBA in the cooked sausages, were positively correlated, probably because these compounds were formed as the products of the nitrosation reaction. Furthermore, NDMA and NPIP were positively correlated with protein, NDMA was negatively correlated with pH, and NPIP was negatively correlated with fat in the fermented sausages. Meanwhile, such correlations were not observed in the cooked sausages. This result might be attributed to the different processing technologies. Although abundant NDMA was produced in lipid-rich conditions [26], amines were indispensable to the formation of NDMA. The abundant protein in fermented sausages might proceed via a degradation step to generate more amines that participated in nitrosation reaction, which tended to occur under acidic conditions, and accounted for the positive and negative correlations with protein and pH, respectively [23]. NDMA was also negatively correlated with 3-DG and positively correlated with GO in the fermented sausages, and NDPhA had a significant negative correlation with DA in the cooked sausages. NAs were not observed to be significantly correlated with AGEs in the fermented and cooked sausages.

## 4. Conclusions

This study investigated the levels of AGEs, NAs, and other indices (α-DCs, moisture, protein, fat, pH, and TBARS) in commercial fermented and cooked sausages, and the correlations between them were discussed. The fermented and cooked sausages were highly different in protein/fat contents and pH/TBARS values due to different processing technologies and added ingredients. AGEs and NAs were not observed to be significantly correlated in the two sausage types. The concentrations of some hazardous compounds, such as CML, NDMA, and NPIP, were observed to be higher in the fermented sausages than in the cooked sausages. Correlation analysis suggested the formation of AGEs could primarily be associated with lipid oxidation in the fermented sausages and with the Maillard reaction in the cooked sausages. With regard to NAs, the levels of NDMA and NPIP were significantly higher in the fermented sausages than in the cooked sausages. NDMA and NDEA were the two primary NAs in all collected sausages, and NPIP was detected only in the fermented sausages. The total NAs levels in some sausage samples exceeded the limit of 10 µg/kg issued by the USDA, suggesting more attention should be paid to mitigating NAs, especially in fermented sausages. These results provided a comprehensive perspective on occurrence of NAs and AGEs in sausages.

Therefore, it would be essential to research effective strategies for controlling these harmful compounds, such as adding antioxidants during sausages processing [1] and focusing more on processing and storage conditions [17]. Moreover, the detailed kinetics of AGEs and NAs formation in sausages also need further exploration.

## Figures and Tables

**Figure 1 foods-12-00724-f001:**
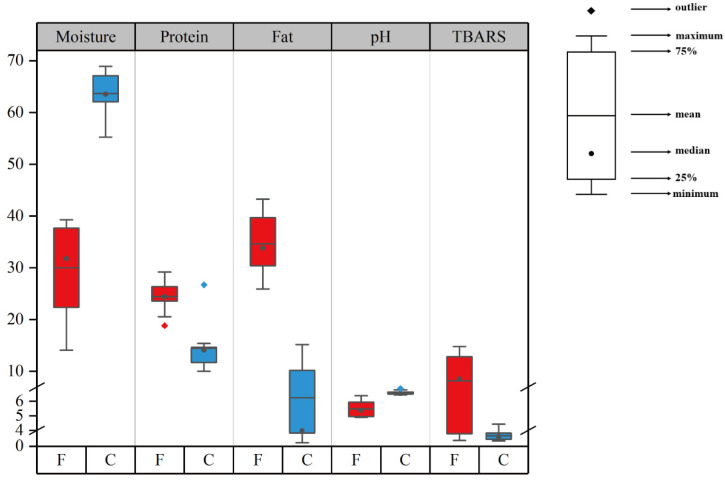
Moisture, protein, fat, pH, and thiobarbituric acid reactive substance (TBARS) contents in the two categories of sausages. F—fermented sausages; C—cooked sausages. The units of moisture, fat, and protein are g/100g, and the unit of TBARS is mg/kg.

**Figure 2 foods-12-00724-f002:**
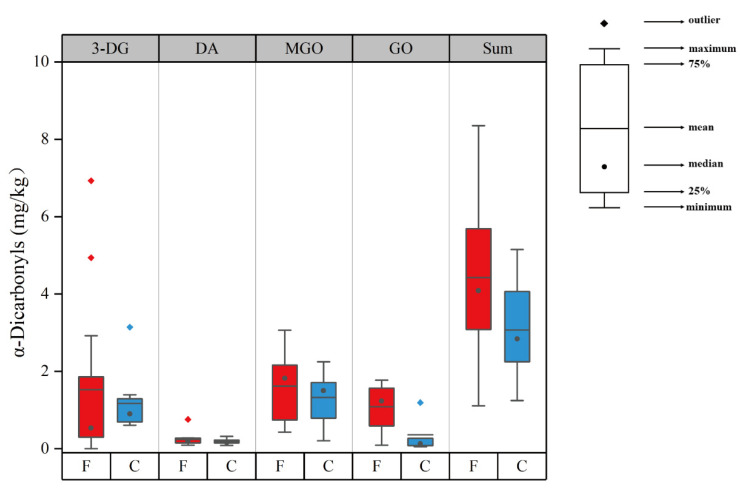
3-DG, DA, GO, and MGO contents in the two different categories of sausages. 3-DG—3-deoxyglucosone; DA—diacetyl; GO—glyoxal; MGO—methylglyoxal; Sum—sum of four kinds of α-dicarbonyls; F—fermented sausages; C—cooked sausages.

**Figure 3 foods-12-00724-f003:**
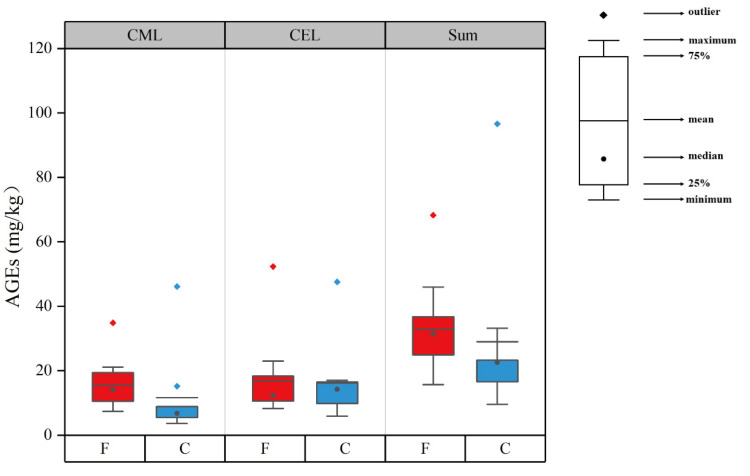
CML and CEL contents of the two categories of sausages. CML—N^ε^-carboxymethyllysine; CEL—N^ε^-carboxyethyllysine; AGEs—advanced glycation end-products; F—fermented sausages; C—cooked sausages; Sum—sum of CML and CEL contents.

**Figure 4 foods-12-00724-f004:**
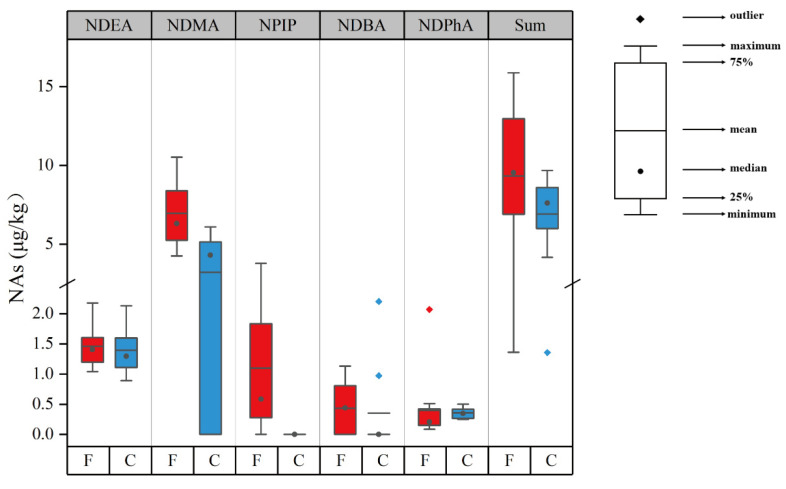
NAs contents of the two categories of sausages. NDMA—N-nitrosodimethylamine; NDEA—N-nitrosodiethylamine; NPIP—N-nitrosopiperidine; NPYR—N-nitrosopyrrolidine; NDBA—N-nitrosodibutylamine; NDPhA—N-nitrosodiphenylamine; Sum—sum of detected N-nitrosamines; F—fermented sausages; C—cooked sausages.

**Figure 5 foods-12-00724-f005:**
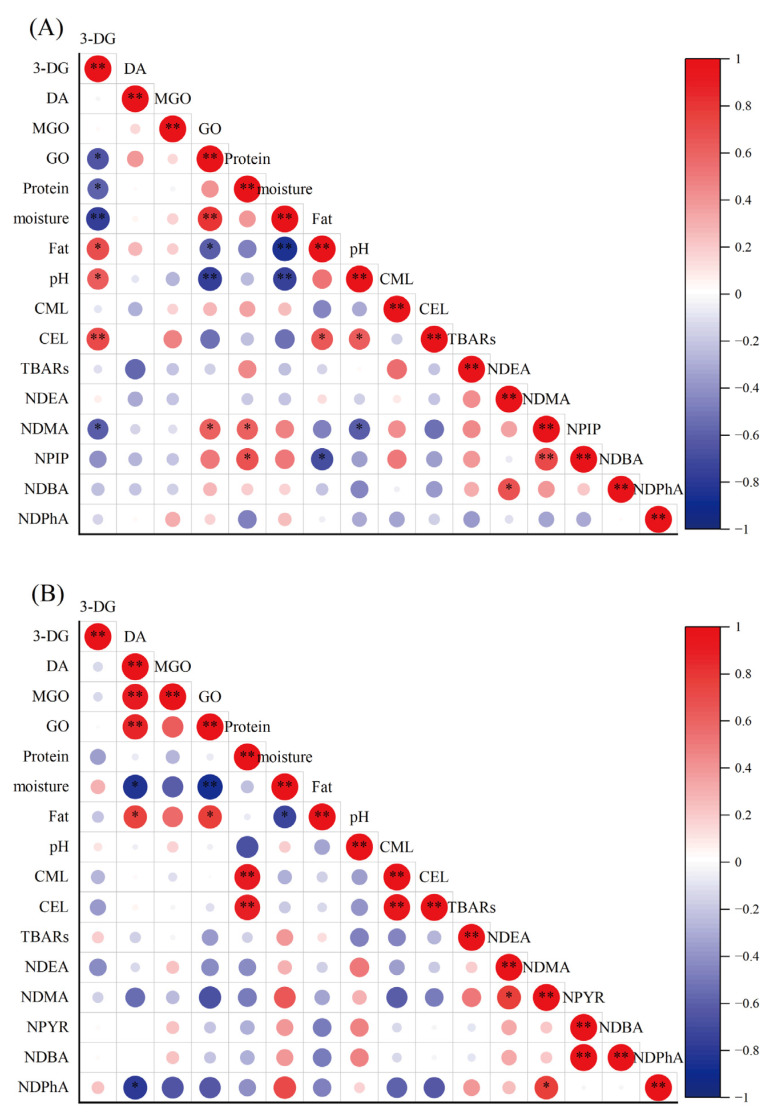
Pearson correlation coefficients between the evaluated indices detected in fermented sausages (**A**) and cooked sausages (**B**). Red and blue colors represent correlation coefficients from 1 to −1, and the size of the circles represents the degree of significance; *: the correlation coefficient is significant at *p* < 0.05; **: the correlation coefficient is significant at *p* < 0.01.

## Data Availability

Not applicable.

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
