# Peer review of "Investigation on the Contents of Nε-carboxymethyllysine, Nε-carboxyethyllysine, and N-nitrosamines in Commercial Sausages on the Chinese Market"

_foods, 2023, doi:10.3390/foods12040724_

Round 1

Reviewer 1 Report

The paper entitled „Investigation on the contents of Nε-carboxymethyllysine, Nε-carboxyethyllysine, and N-nitrosamines in commercial sausages on the Chinese market” presents a simultaneous investigation of AGEs (CML, CEL), α-DCs (3-DG, DA, MGO, GO), and NAs (NDEA, NDMA, NPIP, NDBA, NDPhA) in fermented and cooked sausages. The manuscript is interesting and valuable due to the amount of examined parameters, however in my opinion it may be further improved.

A. The major suggestions:

1. Authors mention 10 samples of cooked sausages in chapter 2.2. However there are only 9 samples presented in the tables S2-S4!

2. There is no information about statistic tests used to compare concentrations of analyzed compounds in the two types of sausages. There is only one test used for correlation evaluation mentioned. It is enough to mention the significance level (p) only once.

3. In my opinion the manuscript should be checked by a native speaker to ensure there are no syntax, spelling or grammar mistakes.

B. The minor suggestions:

1. There are double spacebars (lines 20, 52, 264, and Table S2), lack of spacebar (136, 188) and some editing errors (9, 247, 578, Table S4 and sample C6) in the text.

2. I suggest use of “advanced glycation end-products” instead of “advanced glycation end products”.

3. Some abbreviations are not explained the first time they appear in the paper: TBARs and USDA in abstract, HPLC in lines 78 and 79, LC-MS/MS in line 118, TCA in line 153, GC-MS/MS in line 177. LOQ should be added in the line 130.

4. It is Maillard reaction, not Millard reaction!

5. Could the authors explain more clearly in the introduction why specifically dietary CML and CEL are considered harmful? There is a difference between dietary and endogenous AGEs.

6. Please check if 12 M HCl (line 108) is not a mistake. Additionally in the same line, please add “the samples were hydrolyzed overnight”.

7. Line 116, “elute” should be changed to “eluate”.

8. Line 144, there is no such chapter as 2.4.2. in the paper.

9. The supplementary Tables and Figures have incorrect numbers. There are Table 1 and Table 2 instead of Table S1 and Table S2. There is Figure 1 instead of Figure S1.

10. The equation in line 158 should be written using Word equations tool.

11. Line 173, I suggest “and samples were centrifuged” instead of “and were centrifuged”.

12. There are some English language mistakes, for example it is written ramp instead of ramped (line 183), TRARs instead of TBARs (line 246), are instead of is (line 289), were instead of was (line 383), marke instead of market (line 417). I noticed lack of commas in some places.

13. Data presented in lines 189-193 should be placed in a table.

14. Figure S1 A is not commented in the paper.

15. I suggest authors to comment on the content of Figure 1 in an order of indexes presented in the figure, e.g. firstly moisture, then protein, then fat, etc.

16. I suggest the sentence “The manufacturing of fermented sausages is involved in fermentation and long days of ripening, which contribute to lower moisture content, higher protein and fat content compared to the cooked sausages that are produced by heat treatment and short processing time.” to be changed to “The fermentation and long days of ripening are involved in manufacturing of fermented sausages, which contributes to lower moisture content, higher protein and fat content compared to the cooked sausages that are produced by heat treatment and short processing time.”

17. I suggest to add a pictorial legend to Figures 1-4 instead of a description in text.

18. Be more consistent while referencing figures, please. In the paper sometimes it is written “Figure”, sometimes “Fig”, and sometimes “Fig.”.

19. Line 269, there should be “two sausage types” instead of “two sausages”.

20. Line 277, glyoxal and glycerol are not the same molecules!

21. Lines 300-302, the sentence is hard to understand.

22. Line 302, p of statistical significance is not presented.

23. Line 303, I think the term “total AGEs” should not be used here, as total content of AGEs can be analyzed by various methods. In the case of this paper it is only a sum of CML and CEL concentrations, and there are many more AGEs.

24. Use either μg/kg or ng/g consistently in text and tables when referring to NAs.

25. Line 319, there is no such number as 10.53 in the Table S4 for NDMA.

26. Lines 349-350, check the abbreviations again.

27. Lines 405-406, it should be mentioned that these concentration were higher in fermented sausages than in cooked sausages.

28. Line 429, should it be bold or not?

29. Table S1, * is not explained in a footnote.

30. Tables S3 and S4, nd is not explained in a footnote.

31. I suggest adding average concentrations for fermented and cooked sausages in Tables S3 and S4.

32. The quality of Figure S1 is very low. Every font has a different size and type. The ranges visible on the lower chromatogram (1-1, 2-2, etc.) are not explained. In Figure S1 A the X axis is labelled, but in the S1 B it is not. In the description it should be “advanced” instead of “Advanced” and “N-nitrosamines” instead of “N-nitrosamine”.

Author Response

Thank you very much for your careful review and professional suggestions.

Reviewer 2 Report

The manuscript entitled "Investigation on the contents of Nε -carboxymethyllysine, Nε - 2 carboxyethyllysine, and N-nitrosamines in commercial sau- 3 sages on the Chinese market" has a high research interest reading. 

I only have two suggestions: 

1) It would be of interest to write more information regarding consequences on health of these molecules (Line 39-41). 

2) It would be important to mention what type of studies could be performed in the future 

3) Shelf life has an important role in the development of these molecules, and in this study, all of sausages were bought in the market, so possibly the amount of them differ. How can this variable be controlled? 

Author Response

Thank you very much for your professional suggestions.

Round 2

Reviewer 1 Report

Dear Authors, Thank you for your correction and comments.

Author Response

Dear Reviewer:

Thank you very much for your professional suggestion.